# Polyneuritis Cranialis Associated with BNT162b2 mRNA COVID-19 Vaccine in a Healthy Adolescent

**DOI:** 10.3390/vaccines10010134

**Published:** 2022-01-17

**Authors:** Pimchanok Kulsirichawaroj, Oranee Sanmaneechai, Orasri Wittawatmongkol, Kulkanya Chokephaibulkit

**Affiliations:** Department of Pediatrics, Faculty of Medicine Siriraj Hospital, Mahidol University, Bangkok 10700, Thailand; pimchanok.kul@mahidol.ac.th (P.K.); orasriw@hotmail.com (O.W.); kulkanya.cho@mahidol.ac.th (K.C.)

**Keywords:** BNT162b2, COVID-19 vaccination, multiple cranial neuropathies, polyneuritis cranialis

## Abstract

A 16-year-old Thai girl developed right facial palsy, a lower motor neuron lesion, and numbness 3 h after receiving the first dose of the BNT162b2 mRNA vaccine. Neurological examination showed the involvement of the right cranial nerves (CN) V, VII, IX, and X. Electrophysiological tests revealed the absence of an F wave response, suggesting a proximal demyelinating process. Magnetic resonance imaging of the brain demonstrated abnormal enhancement of the right CN VII. The cerebrospinal fluid profile on day 7 after the onset of symptoms was normal. The patient was diagnosed with polyneuritis cranialis, a rare variant of Guillain-Barre syndrome. She was successfully treated with intravenous immunoglobulin therapy.

## 1. Introduction

Coronavirus disease 2019 (COVID-19) is caused by severe acute respiratory syndrome coronavirus 2 (SARS-CoV-2). The disease was first identified during an outbreak in Wuhan, China, and has emerged explosively worldwide. Over approximately 2 years, more than 290 million cases have been reported globally, with more than 5 million total deaths [1]. A lack of prior immunity and unique structural features [2] allow the virus to efficiently infect and transmit to human hosts.

As a result of the COVID-19 pandemic, the US Food and Drug Administration (FDA) issued an Emergency Use Authorization (EUA) allowing the emergency use of an unapproved product, the BNT162b2 (Pfizer-BioNtech) COVID-19 vaccine, for active immunization to prevent COVID-19 in individuals 12 years of age and older [3].

Although the vaccine is generally safe, cases have been reported of temporal association between the vaccine and Bell’s palsy and Guillain-Barre syndrome (GBS) [4,5]. Onsets of one or the other of the two syndromes were reported to occur 1 to 42 days after vaccination. Here, we report the case of a Thai adolescent who developed polyneuritis cranialis 3 h after a BNT162b2 vaccination.

## 2. Case Presentation

A healthy 16-year-old Thai girl with no past history of neurological or infectious disease presented with numbness and drooping on the right side of her face. She also had no past history of adverse reactions to any drug or vaccine. Her symptoms started 3 h after receiving the first dose of BNT162b2 mRNA vaccine intramuscularly in her left deltoid. She described having right face numbness and drooling from the right side of her mouth. The next day, incomplete right eye closure and difficulty furrowing her right eyebrow were observed. Her symptoms rapidly progressed to the disability of all facial expressions and loss of taste.

The patient was admitted to the hospital on day 5 after the onset of the symptoms. On physical examination, she was afebrile and hemodynamically stable. She had hypoesthesia of her right face and complete right facial palsy, indicating lower motor neuron involvement (House–Brackmann facial nerve grade VI; Figure 1), reduced taste sensation at the right anterior two-thirds of her tongue, and the absence of a gag reflex on the right side of her throat. No weakness or numbness of the extremities, areflexia, evidence of meningism, evidence of other neurological deficits, or systemic symptoms were observed.

On the sixth day after the onset of the symptoms, she complained of pain under the right ear. There was a point of tenderness just below the right temporomandibular joint. Physical examination revealed palpable swelling of the right facial nerve.

Complete blood count, serum electrolyte, thyroid function test, venereal disease research laboratory (VDRL) tests, and anti-nuclear antibody (ANA) were normal. Negative results were obtained from nasopharyngeal swabs for reverse transcription-polymerase chain reaction to test for severe acute respiratory syndrome coronavirus 2 (SARS-CoV-2) and multiplex PCR for respiratory virus panels. An electrophysiological study using Nicolet Viking version 20.1.30 Middleton, WI, USA revealed right facial neuropathy (Figure 2) and the absence of F wave responses suggestive of a proximal demyelinating process (Figure 3). Post-gadolinium T1-weighted imaging of GE signa 3-tesla magnetic resonance images (MRI) of the brain and cranial nerve revealed abnormal enhancement of the right CN VII, which is compatible with neuritis (Figure 4). An unexpected finding was the discovery of an arachnoid cyst at the right cerebellopontine angle, which caused stretching of the right CN VII and VIII and the IX-XI complex. A lumbar puncture was performed, and the cerebrospinal fluid profile was normal, including protein level and cell count (WBC 1/mm^3^; protein 13 mg/dL). Anti-gangliosides GM1, GM2, GM3, GD1a, GD1b, GT1b, GQ1b IgM, and IgG were negative.

The patient was diagnosed with polyneuritis cranialis, a rare variant of GBS, which was temporally associated with the BNT162b2 vaccination. She was successfully treated with intravenous immunoglobulin (IVIG) 2 g/kg. It was divided equally and administered over 2 days. Her symptoms gradually improved. Decreased levels of right facial paresthesia and return of the right gag reflex were observed on the second and third days after the IVIG, respectively. Her hospital stay was uncomplicated, and she was discharged on the ninth day of admission. After discharge, she reported normal sensations on the right size of her face and normal taste at 1 and 2 weeks, respectively. The pain in the temporomandibular joint resolved 3 weeks after discharge. At her 4-week follow-up, all previous neurological deficits were found to be resolved, except for the remaining right facial palsy (House–Brackmann facial nerve grade III).

## 3. Discussion

The term polyneuritis cranialis has been used to describe patients with multiple cranial neuropathies in the absence of weakness of the limbs. The condition is a rare variant of GBS, and its incidence is unknown. Since there are no clear diagnostic criteria, diagnosis is usually made by clinical presentation combined with albuminocytological dissociation in CSF, neurophysiological evidence of neuropathy, or the presence of anti-GQ1b or anti-GT1a IgG antibodies [6].

The diagnosis of polyneuritis cranialis in our patient was based on clinical manifestations. The finding of absent F waves in the electrophysiological study, which is the earliest sign for identifying proximal demyelination, suggested the diagnosis of a GBS variant. The presence of the arachnoid cyst at the right cerebellopontine angle was probably an incidental finding due to the reversal of symptoms after treatment despite the persistence of the cyst.

Polyneuritis cranialis was previously reported in an adult who received the AstraZeneca COVID-19 vaccine [5]. Polyneuritis cranialis was also reported as a manifestation of SARS-CoV-2 infection [7,8]. SARS-CoV-2-associated polyneuritis cranialis may involve not only cranial nerves III, IV, and VI but also other cranial nerves and peripheral nerves [8]. GBS has been reported in 19 patients who received a COVID-19 vaccine from Pfizer, AstraZeneca, or Johnson & Johnson [5]. We report the first case of polyneuritis cranialis associated with the BNT162b2 (Pfizer-BioNtech) COVID-19 vaccine in an adolescent. In our patient, infectious causes associated with GBS variants were excluded by negative nasopharyngeal swabs for a reverse transcription-polymerase chain reaction to test for SARS-CoV-2 and multiplex PCR for respiratory panels.

Postvaccination neuropathy has been reported previously [9]. Peripheral nerve biopsies performed on three patients who had neuropathies after vaccination confirmed the presence of inflammatory deposits in the endoneurium. The exact mechanisms are not yet clearly understood. In the case of our patient, her rapid clinical response to the IVIG suggests that immune-mediated inflammatory demyelination of the cranial nerves was the most likely mechanism.

The rapid development of multiple cranial neuropathies shortly after vaccination in our patient suggested an association between the vaccine and her symptoms and signs. However, a vaccine-induced immune-mediated response usually takes days to develop. Consequently, the cause of this condition in our patient remains unclear. Patone et al. recently conducted a self-controlled case series study to investigate neurological complications, including GBS, within 28 days after the first dose of the AstraZeneca or Pfizer COVID-19 vaccines. They found an increased risk of neurological complications in people who received the COVID-19 vaccine, with a higher risk after a positive SARS-CoV-2 test [10]. They concluded that the risks associated with COVID-19 vaccination are far lower than those associated with COVID-19 infection.

## 4. Conclusions

Since mass vaccination campaigns are being conducted worldwide to control the outbreak of SARS-CoV-2, physicians should be aware of neurological complications that can be temporally associated with the vaccine. We present the first case of polyneuritis cranialis, a variant of GBS, that developed 3 h after the administration of the first dose of the BNT162b2 vaccine in a Thai adolescent. However, this adverse event remains rare, and the overall risk of neurological complications remains low.

## Figures and Tables

**Figure 1 vaccines-10-00134-f001:**
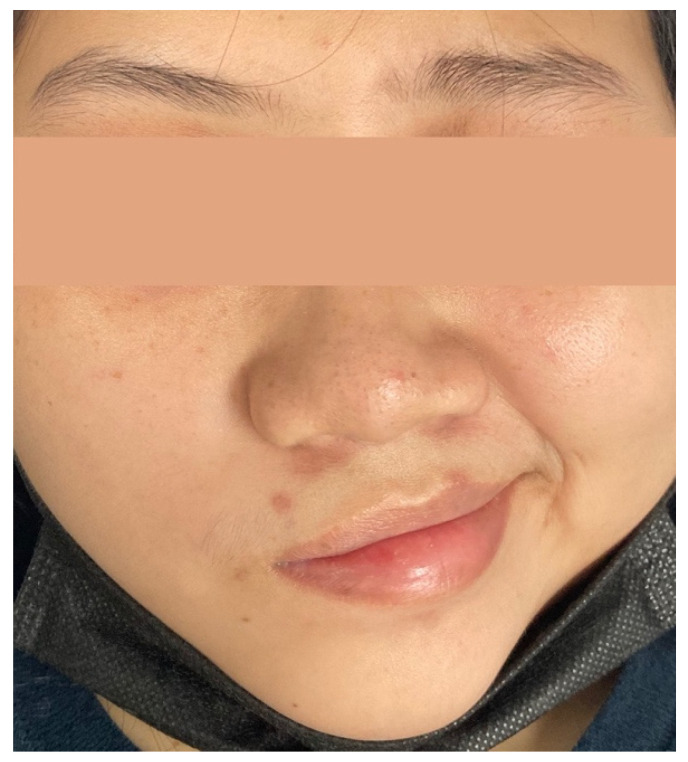
Image of the neurological status of the patient on admission. Multiple cranial nerve (CN) palsies (right CN V, VII, IX, and X); asymmetrical eyebrows; incomplete right eye closure; and complete loss of right nasolabial fold (House–Brackmann facial nerve grade VI) can be observed.

**Figure 2 vaccines-10-00134-f002:**
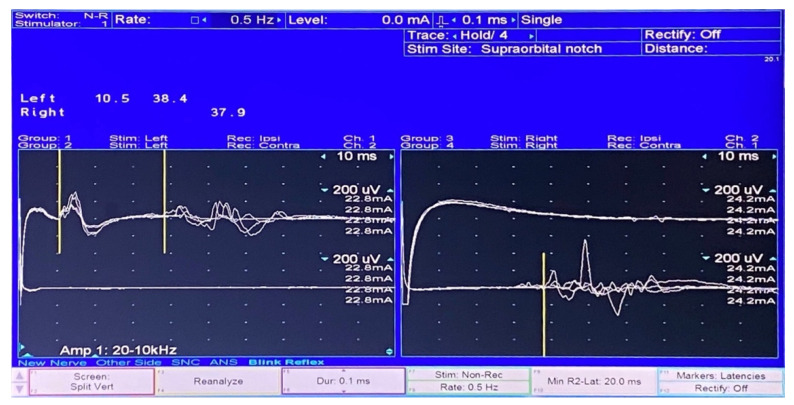
The blink reflex study showed the absent right R1 and R2 when stimulating the right side. Contralateral R2 responses were normal (**right**). The left R1 response was normal when stimulating the left side. Ipsilateral and contralateral R2 responses were normal (**left**).

**Figure 3 vaccines-10-00134-f003:**
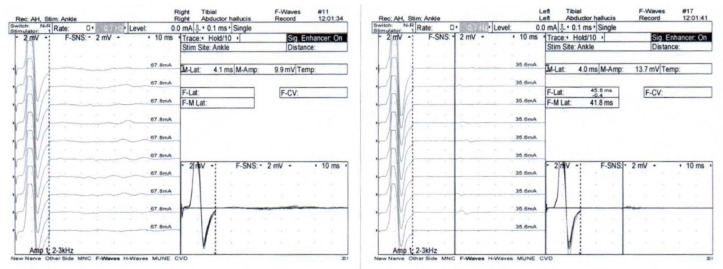
Absence of an F wave response of the right and left tibial nerves was demonstrated.

**Figure 4 vaccines-10-00134-f004:**
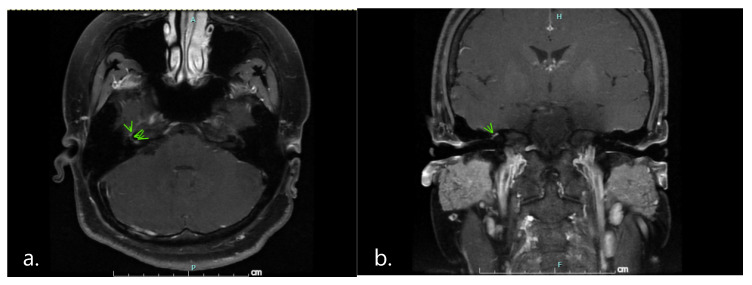
Post-gadolinium T1-weighted (**a**) an axial view; (**b**) a coronal view imaging of magnetic resonance images revealed abnormal enhancement of right cranial nerve VII at the canalicular and labyrinthine segments and the genu (arrowheads). These findings are compatible with neuritis.

## Data Availability

Not applicable.

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
