# Peer review of "Polyneuritis Cranialis Associated with BNT162b2 mRNA COVID-19 Vaccine in a Healthy Adolescent"

_vaccines, 2022, doi:10.3390/vaccines10010134_

Round 1
Reviewer 1 Report
Kulsirichawaroj et al., reported a case of Polyneuritis Cranialis three hours after 16 years old Thia girl received an mRNA COVID-19 vaccine. The interesting case as it contribute to the growing literature of understanding the effects of the mRNA COVID-19 on humans.
Major comments.
The introduction must be improved.
The case report section and discussions must be improved to enhance the readability and understanding of the manuscript.
The manuscript needs serious English editing.
The authors should add details of their methods (i.e., Neurological, Magnetic resonance imaging). I suggest a method section should be included.
Below are some comments on the manuscript
The title should go “Polyneuritis Cranialis Associated with BNT162b2 mRNA COVID-19 Vaccine”.
Line 11….Thai girl….please correct this
Line 12 BNT162b2 mRNA vaccine
Line 14 please replace “suggestive of” with “suggesting” or indicating
Line 17 please delete “she”
Line 20 please improve the introduction by adding the causive agent of COVID 19 and possible where it was fest detected. Also add the infection death rate of COVID 19 (https://doi.org/10.1111/febs.15651 ; https://coronavirus.jhu.edu/map.html ). Also define COVID 19 in full and the causive agent (SARS-CoV-2)Line 29 please could you be specific about the “days or weeks”
line 33 delete “previously”.
Author Response
We thank the reviewers for all their comments and recommendations. We rewrite our manuscript to meet the requirements of the Reviewers. The addition of information enriched our manuscript. According to reviewer’s suggestion, we have decided to change the title to ‘Polyneuritis Cranialis Associated with BNT162b2 mRNA COVID-19 Vaccine in a Healthy Adolescent’.
Responses to the 1st Reviewer comments:
- The introduction must be improved. Improve the introduction by adding the causive agent of COVID 19 and possible where it was fest detected. Also add the infection death rate of COVID 19 (https://doi.org/10.1111/febs.15651; https://coronavirus.jhu.edu/map.html ). Also define COVID 19 in full and the causive agent (SARS-CoV-2)
Author response:
Thank you for your advice. We improved the introduction to make it more readable. As you suggested, we added the causive agent of COVID 19 and number of total death rate worldwide in the first paragraph of introduction and also added reference 1 and 2.
- The case report section and discussions must be improved to enhance the readability and understanding of the manuscript.
Author response:
We revised the case presentation to enhance the readability and adding the discussion of arachnoid cyst which should be an incidental finding because of the reversal of symptom after treatment.
- The manuscript needs serious English editing.
Author response:
We improved the text with English editing proven by a native English speaker that specializes in editing scientific research written by researchers whose native language is not English.
- The authors should add details of their methods (i.e., Neurological, Magnetic resonance imaging). I suggest a method section should be included.
Author response:
We add some details of some investigations described in the fourth paragraph of “2. Case Presentation”.
- The title should go “Polyneuritis Cranialis Associated with BNT162b2 mRNA COVID-19 Vaccine”.
Author response:
Thank you for your suggestion. We revised the title as your recommendation from ‘Polyneuritis Cranialis Temporally Associated with BNT162b2 COVID-19 Vaccination in a Healthy Adolescent’ to ‘Polyneuritis Cranialis Associated with BNT162b2 mRNA COVID-19 Vaccine in a Healthy Adolescent’. However, we left “in a Healthy Adolescent” at the end because we would like to emphasize that this event occurred in this age group, never been described in previous publications.
- Line 11….Thai girl….please correct this
Line 12 BNT162b2 mRNA vaccine
Line 14 please replace “suggestive of” with “suggesting” or indicating
Line 17 please delete “she”
Line 33 delete “previously”
Author response:
Those have been corrected in the text. - Line 29 please could you be specific about the “days or weeks”
Author response:
We added the range of symptom onset following COVID 19 vaccination as described in reference 4-5.
Reviewer 2 Report
This is a case report about the development of a rare variant of Guillain-Barre syndrome (GBS), temporally associated with BNT162b2 vaccination in a teenage girl.
Importantly, the manuscript reports the successful treatment with intravenous immunoglobulin, and complete reversal of all symptoms, except for a remaining right facial palsy, on a 4-week follow-up, highlighting that the risks associated with COVID-19 vaccination are far lower than the risks associated with COVID-19 infection.
The case report is well documented and all relevant clinical information is presented. As acknowledged by the authors, polyneuritis cranialis was already reported as a manifestation of SARS-CoV-2 infection, but - as a minor point to review - a more comprehensive and recent reference could be used (see the more recent manuscript by Finsterer, Environ Sci Pollut Res Int. 2021 Jun 11 : 1–3. doi: 10.1007/s11356-021-14797-3).
Author Response
We thank the reviewers for all their comments and recommendations. We rewrite our manuscript to meet the requirements of the Reviewers. The addition of information enriched our manuscript. According to reviewer’s suggestion, we have decided to change the title to ‘Polyneuritis Cranialis Associated with BNT162b2 mRNA COVID-19 Vaccine a Healthy Adolescent’.
Responses to the 2nd Reviewer comments:
- As acknowledged by the authors, polyneuritis cranialis was already reported as a manifestation of SARS-CoV-2 infection, but - as a minor point to review - a more comprehensive and recent reference could be used (see the more recent manuscript by Finsterer, Environ Sci Pollut Res Int. 2021 Jun 11 : 1–3. doi: 10.1007/s11356-021-14797-3).
Author response:
Thank you for your suggestion. We reviewed this reference and included more information in the Discussion section.
Reviewer 3 Report
- I wonder if you should add the House Brackmann facial nerve grade 6 to the legend in the figure 1?
- Although the patient's eyes are covered and she has given consent for this publication, it might be useful to add more strips to cover the upper forehead and hair to limit identification of the patient.
Author Response
We thank the reviewers for all their comments and recommendations. We rewrite our manuscript to meet the requirements of the Reviewers. The addition of information enriched our manuscript. According to reviewer’s suggestion, we have decided to change the title to ‘Polyneuritis Cranialis Associated with BNT162b2 mRNA COVID-19 Vaccine a Healthy Adolescent’.
Responses to the 3nd Reviewer comments:
- I wonder if you should add the House Brackmann facial nerve grade 6 to the legend in the figure 1?
Author response:
We added as your suggestion.
- Although the patient's eyes are covered and she has given consent for this publication, it might be useful to add more strips to cover the upper forehead and hair to limit identification of the patient
Author response:
We edited this picture as you advised.
Reviewer 4 Report
Thanks to the Authors to present this intersting case during the pandemic.
However there are some concerns about the case:
1-Because there was not any dissociation in protein and cells of the CSF the title of polyneuritis cranialis as subtype of GBS should be change to: A case resembling to or similiar to polyneuritis cranilais type of GBS...
2-The introduction should be writen more precisely.
3-The authors mentioned the presence of a CP angle arachnoid cyst in RT side,however it is was not clear in imaging and also in the development of the signs.Any discussion was explained.
4-The development of the symptoms within hours and resolving within hours after IVIg treatment are vague and they are not be expected scientifically at all,however with Covid-19 or its vaccines we expect anything in the clinics.!
Author Response
"Please see the attachment."

Round 2
Reviewer 1 Report
The authors have revised the manuscript to my satisfaction. Therefore please accept in the present form.